

# Area law and OPE blocks in conformal field theory

Jiang Long$^\star$

School of Physics, Huazhong University of Science and Technology,
Wuhan, Hubei 430074, China

$\star$ longjiang@hust.edu.cn

## Abstract

**This is an introduction to the relationship between area law and OPE blocks in conformal field theory.**

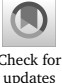
# 1 Introduction

This report consists a summary of our recent progress on the relationship between area law and OPE blocks. Area law has been a continuous topic in physics. The prototype of area law dates back to black hole physics in general relativity. The unusual property that the thermal entropy of a black hole is proportional to the event horizon of the black hole [1, 2] has stimulated various modern idea of theoretical physics, including the famous holographic principle.

OPE block [3, 4], on the other hand, is a relatively unexplored topic in conformal field theory, though it has been defined and discussed at the early stages of conformal field theory [5, 6]. The operator product expansion of two primary operators is equivalent to a summation of OPE blocks with corresponding three point function coefficients. It is a smeared operator which is generated from the so-called (quasi-)primary operator, and extends the study of local operators in CFT to non-local operators.

Modular Hamiltonian, the logarithm of the reduced density matrix [7], plays a central role in the context of geometric entanglement entropy [8–11]. Entanglement entropy is a von Neumann entropy generated from the reduced density matrix of a subregion of spacetime. It suffers divergent problem in general. One can introduce a UV cutoff to secure this problem. An intriguing fact of the entanglement entropy is that it obeys area law in the leading order of the divergences. Its connection to gravity has been established by the work of Ryu and Takayanagi [12], in which they proposed that the entanglement entropy of a CFT is equal to the area of a minimal surface in the bulk AdS spacetime.

On the CFT side, the OPE block provides a novel look at the modular Hamiltonian. Modular Hamiltonian is a special OPE block generated by the stress energy-momentum tensor for a ball region. As we will show, modular Hamiltonian is related to "area laws" in the context of entanglement entropy [1]. This leads to the conjecture that similar to the modular Hamiltonian, general OPE blocks may exhibit area law. Indeed, in a series of papers [13, 14, 16, 17], we have shown that the quantity which satisfies area law is the type-$(m)$ connected correlation function (CCF). More explicitly, the leading term of the type-$(m)$ CCF is proportional to the area of the boundary of the ball. In the subleading terms, we find a logarithmic divergence with degree $q$. In all examples we studied, we found $q = 0, 1, 2$, but in general we don't rule out the possibility of other values. The coefficient $p_q$ for the logarithmic term with degree $q$ is cutoff independent. We establish a relationship between $p_q$ and the type-$(m-1, 1)$ CCF of OPE blocks for two balls which are far away from each other. The coefficient $p_q$ obeys a cyclic identity which is independent of the order of the operators.

This paper is organised as follows. In section 2, we will introduce some basic concepts and conventions used in this paper. Section 3 is devoted to the study of the new area law which is related to the OPE blocks. Various generalizations have been given in section 4. We conclude in section 5 with a number of general open problems that deserve, in our opinion, more work.

# 2 Setup

In this section, we introduce some basic concepts and conventions used in this paper.

---

[1]The " area law" discussed in this paper includes subleading corrections. We use the slogan " area law" following the convention of geometric entanglement entropy.

## 2.1 Area law

In any continues quantum field theory(QFT), physical degrees exist at each point $(t, x^i), i = 1, \cdots, d-1$ of spacetime $M$. At each time slice $t = t_0$, the data on the Cauchy surface $\Sigma$ determines the evolution of the fields. One can divide the surface $\Sigma$ into a spacelike subregion $A$ and its complement $\bar{A}$, $\Sigma = A \cup \bar{A}$. The boundary $\partial A$ is a codimension 2 surface whose area is $\mathcal{A}$. The causal development of A is denoted by $\mathcal{D}(A)$. The physical data on $A$ can only determine the evolution of the fields in $\mathcal{D}(A)$. The causal development $\mathcal{D}(A)$ is an independent subsystem of the original spacetime $M$. Operators in this subsystem are collected to form an algebra $\boldsymbol{a}(A)$. Assume the QFT in the spacetime $M$ is described by a density matrix $\rho$, then by integrating out the degrees of freedom in the complement of $\bar{A}$, one achieves a reduced density matrix $\rho_A$

$$\rho_A = \text{tr}_{\bar{A}} \rho. \tag{2.1}$$

The reduced density matrix $\rho_A$ is a special operator in $\boldsymbol{a}(A)$ since it describes the subsystem $\mathcal{D}(A)$ effectively. A general quantity $\mathcal{Q}(A)$ in $\boldsymbol{a}(A)$ is said to obey area law if its leading term is proportional to the area of the boundary $\partial A$,

$$\mathcal{Q}(A) \propto \mathcal{A} + \cdots. \tag{2.2}$$

The area law defined in (2.2) can be extended to general field theory. One typical example is the black hole entropy in Einstein gravity. The black hole entropy is proportional to the area of its event horizion,

$$S_{bh} = \frac{\mathcal{A}}{4G}, \tag{2.3}$$

where $G$ is the Newton constant. At the loop level, black hole entropy requires logarithmic corrections [18–23]. Usually, the logarithmic correction is in the form $C \log \mathcal{A}$ where the constant $C$ may encode useful information of the black hole.

Sometimes the area law is divergent, one typical example is the geometric entanglement entropy

$$S_A = -\text{tr}_A \rho_A \log \rho_A. \tag{2.4}$$

In this case, one should insert a UV cutoff $\epsilon > 0$,

$$S_A = \gamma \frac{\mathcal{A}}{\epsilon^{d-2}} + \cdots. \tag{2.5}$$

In the subleading terms, there may be a logarithmic term whose coefficient is independent of the cutoff,

$$S_A = \gamma \frac{R^{d-2}}{\epsilon^{d-2}} + \cdots + p \log \frac{R}{\epsilon} + \cdots, \tag{2.6}$$

where the parameter $R$ is the characteristic length of the region $A$.

In this report, we will present a quantity $\mathcal{Q}(A)$ which has a slightly different logarithmic behaviour

$$\mathcal{Q}(A) = \gamma \frac{R^{d-2}}{\epsilon^{d-2}} + \cdots + p_q \log^q \frac{R}{\epsilon} + \cdots. \tag{2.7}$$

The maximum power $q$ of the logarithmic terms is a nonnegative integer. We will call it the degree of the quantity $\mathcal{Q}(A)$. The coefficient $p_q$ is cutoff independent and encodes useful information of the theory. There could be logarithmic pieces with smaller power, however, their coefficients are not universal under a rescaling $\epsilon \to \lambda \epsilon$. In the special case that the subregion $A$ is a ball, $R$ could be chosen to be its radius. The subregion $A$ and its causal development $\mathcal{D}(A)$ are in one-to-one correspondence, we will not distinguish them in the following.

Finally, let's further comment on the area law and logarithmic behaviour studied in this paper.

- In two dimensions, there is no polynomial term of $\frac{R}{\epsilon}$, the modified "area law" is

$$\mathcal{Q}(A) = p_q \log^q \frac{R}{\epsilon} + \cdots. \tag{2.8}$$

- In higher dimensions ($d > 2$), the leading term is always proportional to the area. One should notice that this term is non-universal and the interesting part is the subleading logarithmic term.

## 2.2 OPE block

In any d dimensional CFT, operators are classified into (quasi-)primary operators $\mathcal{O}$ and their descendants $\partial_\mu \partial_\nu \cdots \mathcal{O}$. A general primary operator is characterized by two quantum numbers, conformal weight $\Delta$ and $so(d-1)$ spin $J$. Under a global conformal transformation $x \to x'$, a primary spin 0 operator transforms as

$$\mathcal{O}(x) \to |\frac{\partial x'}{\partial x}|^{-\Delta/d} \mathcal{O}(x), \tag{2.9}$$

where $|\partial x'/\partial x|$ is the Jacobian of the conformal transformation of the coordinates, $\Delta$ is the conformal weight of the primary operator. Operator product expansion(OPE) of two separated primary scalar operators $\mathcal{O}_i(x_1)\mathcal{O}_j(x_2)$ is to expand their product in a local orthogonal and complete basis around a suitable point

$$\mathcal{O}_i(x_1)\mathcal{O}_j(x_2) = \sum_k C_{ijk}|x_{12}|^{\Delta_k - \Delta_i - \Delta_j}(\mathcal{O}_k(x_2) + \cdots), \tag{2.10}$$

where $\cdots$ are descendants of the primary operator $\mathcal{O}_k$. Its form is fixed by global conformal symmetry, therefore it just contains kinematic information of the CFT. The summation is over all possible primary operators of the CFT. Here we expand the product around the point $x_2$. The distance of any two points $x_i, x_j$ is written as $|x_{ij}|$. The constant $C_{ijk}$ is called the OPE coefficient which is related to the three point function of primary operators

$$\langle \mathcal{O}_i(x_1)\mathcal{O}_j(x_2)\mathcal{O}_k(x_3) \rangle = \frac{C_{ijk}}{|x_{12}|^{\Delta_{12,3}}|x_{23}|^{\Delta_{23,1}}|x_{13}|^{\Delta_{13,2}}}, \quad \Delta_{ij,k} = \Delta_i + \Delta_j - \Delta_k. \tag{2.11}$$

They are the only dynamical parameters in the CFT. The constants $\Delta_i, \Delta_j, \Delta_k$ are conformal weights of the corresponding primary operators. By collecting all kinematic terms in the summation, we can rewrite the OPE (2.10) as

$$\mathcal{O}_i(x_1)\mathcal{O}_j(x_2) = |x_{12}|^{-\Delta_i - \Delta_j} \sum_k C_{ijk} Q_k^{ij}(x_1, x_2). \tag{2.12}$$

The objects $Q_k^{ij}(x_1, x_2)$ are called OPE blocks [3,5,6]. They are non-local operators in the CFT and depend on the position $x_1$ and $x_2$ of the external operators. The upper index $i$ and $j$ show that it also depends on the quantum numbers of the external operators $\mathcal{O}_i$ and $\mathcal{O}_j$. It is easy to see that OPE block has dimension zero. Under a global conformal transformation $x \to x'$, an OPE block $Q_k^{ij}(x_1, x_2)$ will transform as

$$Q_k^{ij}(x_1, x_2) \to f(x_1', x_2')Q_k^{ij}(x_1', x_2'). \tag{2.13}$$

The explicit form of $f(x_1', x_2')$ is not important in this work. When the two external operators have the same quantum numbers, we have $f(x_1', x_2') = 1$ and OPE block will be invariant under the global conformal transformation. One can also show that the OPE block is independent

of the external operator in this special case. Due to this reason, we relabel such kind of OPE block as

$$Q_A[\mathcal{O}_k] = Q_k^{ii}(x_1, x_2). \tag{2.14}$$

The subscript $A$ denotes the region determined by the two points $x_1$ and $x_2$ where the two external operators are inserted. The operator in the square bracket reflects the fact that OPE block is generated by a primary operator $\mathcal{O}_k$. We omit the information of $i$ since the OPE block is insensitive to the external operators in this case. We will classify the primary operators $\mathcal{O}_k$ into conserved currents $\mathcal{J}$ and non-conserved operators $\mathcal{O}$. A general symmetric traceless primary operator obeys the following unitary bound [24]

$$\begin{cases} \Delta \geq J + d - 2, & J \geq 1, \\ \Delta \geq \frac{d-2}{2}, & J = 0. \end{cases}$$

A conserved current $\mathcal{J}$ with spin $J(J \geq 1)$ will satisfy $\Delta = J + d - 2$. All other primary operators are non-conserved operators. Correspondingly, the OPE block (2.14) generated by a conserved current $\mathcal{J}$ will be called a type-J OPE block. On the other hand, the OPE block (2.14) generated by a non-conserved operator $\mathcal{O}$ will be called a type-O OPE block.

When two operators are time-like separated, the region $A$ is a causal diamond. The two operators are at the sharp corner of the diamond $A$. We can use the conformal transformation to fix

$$x_1 = (1, \vec{x}_0), \quad x_2 = (-1, \vec{x}_0), \tag{2.15}$$

then the causal diamond $A$ intersects the $t = 0$ slice at a unit ball $(R = 1)$ which we will also denote it as $A$

$$A = \{(0, \vec{x}) | (\vec{x} - \vec{x}_0)^2 \leq 1\}. \tag{2.16}$$

The center of the ball is $\vec{x}_0$. The boundary of the ball $A$ is a unit sphere $\partial A$. In the context of geometric entanglement entropy, the surface $\partial A$ is an entanglement surface which separates the ball $A$ and its complement. The leading term of entanglement entropy is proportional to the area of the surface $\partial A$ in general higher dimensions $(d > 2)$. In two dimensions, the entanglement entropy is logarithmically divergent with the logarithmic degree $q = 1$. There is a conformal Killing vector $K$ which preserves the diamond $A$,

$$K^\mu = \frac{1}{2}(1 - (\vec{x} - \vec{x}_A)^2 - t^2, -2t\vec{x}). \tag{2.17}$$

The conformal Killing vector $K$ is null on the boundary of the diamond $A$. It generates a modular flow of the diamond $A$. A type-O OPE block corresponding to the point pair (2.15) or the unit ball $A$ (2.16) is [4]

$$Q_A[\mathcal{O}_{\mu_1 \cdots \mu_J}] = c_{\mathcal{O}_{\mu_1 \cdots \mu_J}} \int_{\mathcal{D}(A)} d^d x K^{\mu_1} \cdots K^{\mu_J} |K|^{\Delta - d - J} \mathcal{O}_{\mu_1 \cdots \mu_J}, \tag{2.18}$$

where the primary operator $\mathcal{O}_{\mu_1 \cdots \mu_J}$ is non-conserved

$$\partial^{\mu_1} \mathcal{O}_{\mu_1 \cdots \mu_J} \neq 0. \tag{2.19}$$

It has dimension $\Delta$ and spin $J$. When the operator is a conserved current

$$\partial^{\mu_1} \mathcal{J}_{\mu_1 \cdots \mu_J} = 0, \tag{2.20}$$

the corresponding type-J OPE block is

$$Q_A[\mathcal{J}_{\mu_1 \cdots \mu_J}] = c_{\mathcal{J}_{\mu_1 \cdots \mu_J}} \int_A d^{d-1} \vec{x} (K^0)^{J-1} \mathcal{J}_{0 \cdots 0}. \tag{2.21}$$

It can be obtained from (2.18) by using the conservation law (2.20) and reducing it to a lower $d-1$ dimensional integral. The coefficient $c_{\mathcal{J}_{\mu_1\cdots\mu_J}}$ is also redefined at the same time. In (2.18) and (2.21), the coefficients $c_{\mathcal{O}_{\mu_1\cdots\mu_J}}$ and $c_{\mathcal{J}_{\mu_1\cdots\mu_J}}$ are free parameters which are fixed by the normalization of the corresponding operators, we set them to be 1.

## 2.3 Modular Hamiltonian and area law

A very special type-J OPE block is the modular Hamiltonian [7,25] of the ball $A$,

$$H_A = 2\pi \int_A d^{d-1}\vec{x}\, K^0 T_{00} = 2\pi \int_A d^{d-1}\vec{x}\, \frac{1-(\vec{x}-\vec{x}_0)^2}{2} T_{00}(0,\vec{x}). \qquad (2.22)$$

Modular Hamiltonian is the logarithm of the reduced density matrix $\rho_A$

$$H_A = -\log\rho_A. \qquad (2.23)$$

It plays a central role in the context of entanglement entropy,

$$S_A = -\mathrm{tr}_A\rho_A\log\rho_A = \mathrm{tr}_A e^{-H_A}H_A. \qquad (2.24)$$

More generally, the Rényi entanglement entropy

$$S_A^{(n)} = \frac{1}{1-n}\log\mathrm{tr}_A\rho_A^n \qquad (2.25)$$

has been shown to satisfy an area law generally

$$S_A^{(n)} = \gamma(n)\frac{\mathcal{A}}{\epsilon^{d-2}} + \cdots, \qquad (2.26)$$

where $\mathcal{A}$ is the area of the entanglement surface $\partial A$ and $\epsilon$ is a UV cutoff. The constant $\gamma(n)$ is cutoff dependent. The subleading terms $\cdots$ contain a logarithmic term with degree $q=1$ in even dimensions

$$S_A^{(n)} = \gamma(n)\frac{\mathcal{A}}{\epsilon^{d-2}} + \cdots + p_1(n)\log\frac{R}{\epsilon} + \cdots, \qquad (2.27)$$

where we have restored the radius $R$ that was previously set to 1. The area $\mathcal{A}$ is related to the radius $R$ through the power law

$$\mathcal{A} \sim R^{d-2}. \qquad (2.28)$$

The coefficient $p_1(n)$ encodes useful information of the CFT. The relation between modular Hamiltonian and area law motivates the conjecture that OPE block maybe related to area law in a suitable way. We will give the framework to discuss this problem in the following subsection.

## 2.4 Deformed reduced density matrix and connected correlation function

Given a primary operator $\mathcal{O}$ in a ball A, one can always define a corresponding OPE block $Q_A[\mathcal{O}]$. We construct an exponential operator formally [14]

$$\rho_A = e^{-\mu Q_A}, \qquad (2.29)$$

which is still in the subregion $A$. The constant $\mu$ is free. Operators of the form (2.29) is called deformed reduced density matrix. Note we use the same symbol $\rho_A$ to label deformed reduced density matrix. Recall that the modular Hamiltonian is a special OPE block, if one replaces the

OPE block by the modular Hamiltonian (2.29) and set $\mu = 1$, the deformed reduced density matrix becomes the reduced density matrix exactly. We can relax the definition, namely, $Q_A$ in (2.29) could be a linear superposition of several OPE blocks. Note our definition of deformed reduced density matrix is a direction extension of the generalized reduced density matrix in the context of the so-called charged Rényi entropy [15]. In that work, $Q_A$ is a charge which is generated by a $U(1)$ current. The corresponding charged Rényi entropy is holographically dual to the thermal entropy of a charged black hole with hyperbolic horizon. However, in our definition, $Q_A$ is just a general OPE block or their linear superposition. As a naive generalization of Rényi entanglement entropy, we construct the logarithm of the vacuum expectation value of the deformed reduced density matrix,

$$T_A(\mu) = \log\langle\rho_A\rangle = \log\langle e^{-\mu Q_A}\rangle. \tag{2.30}$$

When $Q_A$ is modular Hamiltonian, the above quantity is related to the Rényi entropy for the vacuum state.

However, a direct computation of $T_A(\mu)$ is hard in general. A much more severe problem is that OPE block has no lower bound in general, therefore the definition is not valid for general OPE blocks. To solve this problem, we observe that $T_A(\mu)$ could be expanded for small $\mu$,

$$T_A(\mu) = \sum_{m=1}^{\infty} \frac{(-\mu)^m}{m!}\langle Q_A^m\rangle_c. \tag{2.31}$$

The Taylor expansion coefficient

$$\langle Q_A^m\rangle_c = (-1)^m \frac{\partial^m}{\partial\mu^m}T_A(\mu)|_{\mu\to 0} \tag{2.32}$$

is called Type-(m) connected correlation function (CCF) of the OPE block $Q_A$. For each definite $m$, one can always calculate the corresponding CCF without knowing $T_A(\mu)$. The first few CCFs are

$$\begin{aligned}\langle Q_A^2\rangle_c &= \langle Q_A^2\rangle - \langle Q_A\rangle^2, \\ \langle Q_A^3\rangle_c &= \langle Q_A^3\rangle - 3\langle Q_A^2\rangle\langle Q_A\rangle + 2\langle Q_A\rangle^3.\end{aligned} \tag{2.33}$$

Using CCF, there is no issue of lower bound of the OPE block. The convergence of the summation (2.31) could be a hard problem for general OPE blocks. However, for modular Hamiltonian in two dimensional CFT, one can use the summation to define Rényi entropy. As an application of the concept of CCF, we choose the OPE block as the modular Hamiltonian, then it is easy to show that CCF of modular Hamiltonian $H_A$ satisfies area law with logarithmic degree $q = 1$ in even dimensions,

$$\langle H_A^m\rangle_c = \tilde{\gamma}\frac{\mathcal{A}}{\epsilon^{d-2}} + \cdots + \tilde{p}_1^{(m)}\log\frac{R}{\epsilon} + \cdots, \quad m \geq 1. \tag{2.34}$$

The coefficient $\tilde{p}_1^{(m)}$ is determined from $p_1(n)$ by

$$\tilde{p}_1^{(m)} = (-1)^m\partial_n^m(1-n)p_1(n)|_{n\to 1}. \tag{2.35}$$

There could be multiple spacelike-separated balls $A_1, A_2, \cdots$, each region has associate OPE block $Q_{A_i}$. We insert $m_i$ OPE blocks into region $A_i$, then we can define the corresponding type-Y CCF

$$\langle Q_{A_1}^{m_1} Q_{A_2}^{m_2}\cdots\rangle_c, \tag{2.36}$$

where the Young diagram $Y$ is

$$Y = (m_1, m_2, \cdots), \quad m_1 \geq m_2 \geq \cdots \geq 1. \tag{2.37}$$

The generator of all type-Y CCFs is

$$T_{\cup_i A_i}(\mu_1, \mu_2, \cdots) = \log \frac{\langle e^{-\sum_i \mu_i Q_{A_i}} \rangle}{\prod_i \langle e^{-\mu_i Q_{A_i}} \rangle}. \tag{2.38}$$

When there are only two balls $A$ and $B$, the generator is

$$T_{A \cup B}(\mu_1, \mu_2) = \log \frac{\langle e^{-\mu_1 Q_A - \mu_2 Q_B} \rangle}{\langle e^{-\mu_1 Q_A} \rangle \langle e^{-\mu_2 Q_B} \rangle} = \sum_{m_1 \geq 1, m_2 \geq 1} \frac{(-1)^{m_1 + m_2} \mu_1^{m_1} \mu_2^{m_2}}{m_1! m_2!} \langle Q_A^{m_1} Q_B^{m_2} \rangle_c. \tag{2.39}$$

We parameterize $A$ and $B$ as

$$A = \{(0, \vec{x}) | (\vec{x} - \vec{x}_0)^2 \leq 1\}, \quad B = \{(0, \vec{x}) | \vec{x} \leq R'^2\}. \tag{2.40}$$

There is only one cross ratio

$$\xi = \frac{4R'}{x_0^2 - (1 - R')^2}. \tag{2.41}$$

When the two regions $A$ and $B$ are spacelike-separated, $|x_0| > 1 + R'$, the cross ratio is between 0 and 1,

$$0 < \xi < 1. \tag{2.42}$$

In some cases, it is more convenient to use an equivalent cross ratio

$$\eta = \frac{\xi}{1 - \xi} = \frac{4R'}{x_0^2 - (1 + R')^2}. \tag{2.43}$$

For spacelike-separated regions $A$ and $B$, the range of the cross ratio $\eta$ is

$$0 < \eta < \infty. \tag{2.44}$$

Since the OPE block $Q_A[\mathcal{O}]$ is invariant under conformal transformation, any type-$(m_1, m_2)$ CCF should be a function of cross ratio $\xi$ or $\eta$. Actually the OPE block is an eigenvector of the conformal Casimir

$$[L^2, Q_A[\mathcal{O}]] = C_{\Delta, J} Q_A[\mathcal{O}], \tag{2.45}$$

where $L^2$ is the Casimir operator of the global conformal group. The eigenvalue $C_{\Delta, J}$ is

$$C_{\Delta, J} = -\Delta(\Delta - d) - J(J + d - 2). \tag{2.46}$$

Therefore, any type-$(m - 1, 1)$ CCF should be a conformal block up to a constant [2]

$$\langle Q_A[\mathcal{O}_1] \cdots Q_A[\mathcal{O}_{m-1}] Q_B[\mathcal{O}_m] \rangle_c = D^{(d)}[\mathcal{O}_1, \cdots, \mathcal{O}_m] G^{(d)}_{\Delta_m, J_m}(\eta). \tag{2.47}$$

The subscript $\Delta_m, J_m$ are the conformal weight and spin of the primary operator $\mathcal{O}_m$. The index $(d)$ is used to label the dimension of spacetime. The conformal block $G^{(d)}_{\Delta_m, J_m}(\eta)$ can be constructed explicitly in even dimensions [26, 27]. In this paper, we use the convention that

$$G^{(d)}_{\Delta_m, J_m}(\eta) \to \eta^{\Delta_m}, \quad \eta \to 0. \tag{2.48}$$

---

[2] See Appendix A of [16] for a detailed discussion. For each spherical space, there is a pair of timelike separated points that live on the tips of its causal diamond [3]. Therefore, for two balls, one can use the two pairs of timelike separated points to define the corresponding cross ratio $\eta$.

Therefore the overall constant $D^{(d)}[\mathcal{O}_1, \cdots, \mathcal{O}_m]$ is fixed uniquely. When $A$ and $B$ are far away from each other, the type-$(m-1, 1)$ CCF is dominated by

$$\langle Q_A[\mathcal{O}_1] \cdots Q_A[\mathcal{O}_{m-1}] Q_B[\mathcal{O}_m] \rangle_c \approx D^{(d)}[\mathcal{O}_1, \cdots, \mathcal{O}_m] \eta^{\Delta_m}. \tag{2.49}$$

For $m \geq 2$, the coefficients $D^{(d)}[\mathcal{O}_1, \cdots, \mathcal{O}_m]$ are related to the normalization of the primary operators. For any $m \geq 3$, it also contains dynamical information of the theory. The explicit form of the conformal block can be found in [28]. Any type-$(m_1, m_2)$ CCF with $m_1 \geq m_2 \geq 2$ is not a conformal block .

# 3 Area law

We conjecture that the type-$(m)$ CCF of OPE blocks obeys the following area law

$$\langle Q_A[\mathcal{O}_1] \cdots Q_A[\mathcal{O}_m] \rangle_c = \gamma \frac{R^{d-2}}{\epsilon^{d-2}} + \cdots + p_q \log^q \frac{R}{\epsilon} + \cdots . \tag{3.1}$$

The leading term is proportional to the area of the boundary $\partial A$. We have restored the radius $R$ in the formula to balance the dimension. The small positive constant $\epsilon$ is the UV cutoff which is roughly the distance from the cutoff to the boundary $\partial A$. The constant $\gamma$ depends on the choice of the cutoff and the method of regularization, we will not be interested in its explicit value. The $\cdots$ terms are subleading and cutoff dependent. Therefore we omit their forms. The degree $q$ characterizes the maximal power of the logarithmic terms. The coefficient $p_q$ is invariant under the rescaling of the cutoff, therefore it encodes detail universal information of the theory [3]. When all the OPE blocks are equal to the modular Hamiltonian, the degree $q = 1$ for even dimensions according to (2.34). However, as we will see, $q$ is not necessarily equal to 1 in general. To distinguish different type-$(m)$ CCFs in different dimensions, we write the area law (3.1) more explicitly as

$$\langle Q_A[\mathcal{O}_1] \cdots Q_A[\mathcal{O}_m] \rangle_c = \gamma[\mathcal{O}_1, \cdots, \mathcal{O}_m] \frac{R^{d-2}}{\epsilon^{d-2}} + \cdots + p_q^{(d)}[\mathcal{O}_1, \cdots, \mathcal{O}_m] \log^q \frac{R}{\epsilon} + \cdots . \tag{3.2}$$

## 3.1 Continuation

The two formulas (2.47) and (3.2) are actually related to each other through an analytic continuation. We use the example of the two dimensional modular Hamiltonian to illustrate this relation. For any $\text{CFT}_2$, the modular Hamiltonian can be decomposed into the holomorphic and anti-holomorphic part, we focus on the holomorphic part

$$H_A = -\int_{-1}^{1} dz \frac{1-z^2}{2} T(z + x_0) + c. \tag{3.3}$$

The constant $c$ can be fixed by the normalization condition

$$\text{tr}_A \rho_A = \text{tr}_A e^{-H_A} = 1. \tag{3.4}$$

Its value doesn't affect the type-Y CCF with any $\sum_i m_i \geq 2$. We also used the convention $T(z) = -2\pi T_{zz}$ where the subscript $z$ is the holomorphic coordinate $z = t + x$. The radius of the interval $A$ is 1, we have shifted the variable $z$ such that the dependence of the center $x_0$ is

---

[3] Note the constant $p_q$ also depends on the operator normalization.

in the stress tensor. The modular Hamiltonian of region $B$ can be obtained by setting $x_0 = 0$ and restoring the radius $R'$. The type-$(m-1, 1)$ CCF of the modular Hamiltonian is

$$\langle H_A^{m-1} H_B \rangle_c = D^{(2)}[T_{\mu_1 \nu_1}, \cdots, T_{\mu_m \nu_m}] G_2^{(2)}(\eta). \tag{3.5}$$

The two dimensional conformal block for a chiral operator can be labeled by the conformal weight $h$ of the operator

$$G_h^{(2)}(\eta) = (-\eta)^h {}_2F_1(h, h, 2h, -\eta). \tag{3.6}$$

We can move the interval $A$ to $B$ such that they coincide. In this limit, any type-$(m-1, 1)$ CCF should approach a type-$(m)$ CCF . This is equivalent to setting $\eta \to -1$. We can set $x_0 \to 0$ and then take the limit $R' \to 1$,

$$x_A \to 0, \quad R' = 1 - \epsilon, \quad \epsilon \to 0. \tag{3.7}$$

The cross ratio $\xi \to -\infty$ or $\eta \to -1$ by

$$\xi = -\frac{4(1-\epsilon)}{\epsilon^2} \approx -\frac{4}{\epsilon^2}, \quad \eta = -\frac{4(1-\epsilon)}{(2-\epsilon)^2} \approx -1 + \frac{\epsilon^2}{4}. \tag{3.8}$$

On the right hand side of (3.5), we find a logarithmically divergent term in this limit

$$G_2^{(2)}(\eta) = 12 \log \frac{2}{\epsilon} + \cdots = 12 \log \frac{R}{\epsilon} + \cdots \tag{3.9}$$

The left hand side of (3.5) approaches type-$(m)$ CCF, therefore

$$\langle H_A^m \rangle_c = 12 D^{(2)}[T_{\mu_1 \nu_1}, \cdots, T_{\mu_m \nu_m}] \log \frac{R}{\epsilon} + \cdots. \tag{3.10}$$

We read out the cutoff independent coefficient

$$p_1^{(2)}[T_{\mu_1 \nu_1}, \cdots, T_{\mu_m \nu_m}] = 12 D^{(2)}[T_{\mu_1 \nu_1}, \cdots, T_{\mu_m \nu_m}]. \tag{3.11}$$

The relation (3.11) is a typical UV/IR relation for the modular Hamiltonian. The left hand side is the universal coefficient for $B$ and $A$ coinciding (UV). On the right hand side, the $D$ coefficient characterizes the leading order behaviour of CCF when $B$ and $A$ are far away from each other (IR). They provide equivalent information of the CFT since the constant 12 is completely fixed by conformal symmetry. The continuation of the conformal block can be generalized to higher dimensions. For example, in four dimensions, the conformal block associated with stress tensor becomes divergent as $A$ approaches $B$,

$$G_{4,2}^{(4)} \approx \tilde{\gamma} \frac{R^2}{\epsilon^2} + \cdots - 120 \log \frac{R}{\epsilon} + \cdots. \tag{3.12}$$

The leading term is exactly proportional to the area of the boundary and the logarithmic divergent term also appears in the subleading terms. We can read out the type-$(m)$ CCF of the modular Hamiltonian in four dimensions

$$\langle H_A^m \rangle_c = \gamma \frac{R^2}{\epsilon^2} + \cdots + p_1^{(4)}[T_{\mu_1 \nu_1}, \cdots, T_{\mu_m \nu_m}] \log \frac{R}{\epsilon} + \cdots, \tag{3.13}$$

with

$$p_1^{(4)}[T_{\mu_1 \nu_1}, \cdots, T_{\mu_m \nu_m}] = -120 D^{(4)}[T_{\mu_1 \nu_1}, \cdots, T_{\mu_m \nu_m}]. \tag{3.14}$$

Note we obtain the area law and the logarithmic behaviour of the type-$(m)$ CCF of the modular Hamiltonian without using any knowledge of Rényi entanglement entropy. The method of

analytic continuation can be applied to general dimensions and OPE blocks. A conformal block $G^{(d)}_{\Delta,J}(\xi)$ obeys area law in the limit $\xi \to -\infty$ in even dimensions. It has degree $q = 1$ only for $\Delta = J + d - 2$,

$$G^{(d)}_{\Delta,J}(\xi) = \tilde{\gamma} \frac{R^{d-2}}{\epsilon^{d-2}} + \cdots + E^{(d)}[\Delta,J] \log \frac{R}{\epsilon} + \cdots, \quad \xi \to -\infty. \tag{3.15}$$

This means that type-$(m)$ CCF of type-J OPE blocks may always obey area law with degree $q = 1$, the cutoff independent coefficient is

$$p^{(d)}_q[\mathcal{O}_1, \cdots, \mathcal{O}_m] = E^{(d)}[\mathcal{O}_m] \times D^{(d)}[\mathcal{O}_1, \cdots, \mathcal{O}_m]. \tag{3.16}$$

We have replaced the quantum numbers in E function by the corresponding primary operator. For non-conserved operators, the conformal block $G^{(d)}_{\Delta,J}$ also obeys area law in the limit $\xi \to -\infty$ in even dimension, though it has degree $q = 2$

$$G^{(d)}_{\Delta,J}(\xi) = \tilde{\gamma} \frac{R^{d-2}}{\epsilon^{d-2}} + \cdots + E^{(d)}[\Delta,J] \log^2 \frac{R}{\epsilon} + \cdots, \quad \xi \to -\infty. \tag{3.17}$$

Therefore, type-$(m)$ CCF of type-O OPE blocks obeys area law with degree $q = 2$. We can obtain similar UV/IR relations as (3.16). In odd dimensions, the story is the same. The degree $q$ is 0 for type-$(m)$ CCF of type-J OPE blocks and 1 for type-O OPE blocks.

## 3.2 Kinematic information

The function $E^{(d)}[\mathcal{O}]$ is completely fixed by conformal symmetry. It can be obtained by reading out the coefficient of the logarithmic term with degree $q$. For each fixed quantum number $\Delta$ and $J$, there is a unique number $E^{(d)}[\mathcal{O}]$. For any type-J OPE block in two dimensions, the primary operator $\mathcal{O}$ has dimension $\Delta = J = h$. The conformal block (3.6) has degree $q = 1$ in the limit $\eta \to -1$. The function $E^{(2)}[\mathcal{O}]$ is

$$E^{(2)}[\mathcal{O}] = \frac{2\Gamma(2h)}{\Gamma(h)^2}, \quad \Delta = J = h. \tag{3.18}$$

For type-O OPE block, the primary operator $\mathcal{O}$ has dimension $\Delta = h + \bar{h}$ and spin $J = h - \bar{h}$. The conformal block has degree $q = 2$ in the limit $\eta \to -1$. The function $E^{(2)}[\mathcal{O}]$ is

$$E^{(2)}[\mathcal{O}] = \begin{cases} \frac{2^{4h}\Gamma(h+\frac{1}{2})^2}{\pi\Gamma(h)^2} & J = 0, \ h > 0 \\ -\frac{4^{2h-1}\Gamma(h-\frac{1}{2})\Gamma(h+\frac{1}{2})}{\pi\Gamma(h-1)\Gamma(h)} & J = 1, \ h > 1 \\ \frac{4^{2h-3}(h-2)(h-1)(2h-3)(2h-1)\Gamma(h-\frac{3}{2})^2}{\pi\Gamma(h)^2} & J = 2, \ h > 2 \\ \cdots \end{cases} \tag{3.19}$$

In four dimensions, we also find

$$E^{(4)}[\mathcal{O}] = \begin{cases} 12 & \Delta = 3, \ J = 1 \\ -120 & \Delta = 4, \ J = 2 \\ 840 & \Delta = 5, \ J = 3 \\ \cdots \end{cases} \tag{3.20}$$

for conserved currents and

$$E^{(4)}[\mathcal{O}] = \begin{cases} -\frac{2^{2\Delta-1}\Gamma(\frac{\Delta-1}{2})\Gamma(\frac{\Delta+1}{2})}{\pi\Gamma(\frac{\Delta-2}{2})^2} & \Delta > 1, \ J = 0, \\ \frac{2^{2\Delta-1}\Gamma(\frac{\Delta}{2})\Gamma(\frac{\Delta+2}{2})}{\pi\Gamma(\frac{\Delta-3}{2})\Gamma(\frac{\Delta+1}{2})} & \Delta > 3, \ J = 1, \\ -\frac{4^{\Delta-1}(\Delta-2)\Gamma(\frac{\Delta-3}{2})\Gamma(\frac{\Delta+3}{2})}{\pi\Gamma(\frac{\Delta-4}{2})\Gamma(\frac{\Delta+2}{2})} & \Delta > 4, \ J = 2, \\ \cdots \end{cases} \tag{3.21}$$

for non-conserved operators. In three dimensions, we find

$$
E^{(3)}[\mathcal{O}] = \begin{cases} -\frac{2^{2\Delta-1}(\Delta-1)\Gamma(\Delta-\frac{1}{2})}{\sqrt{\pi}\Gamma(\Delta-1)} & \Delta > \frac{1}{2}, \ J = 0. \\[2ex] \frac{2^{\Delta+1}\Delta\Gamma(\Delta-\frac{1}{2})}{\Gamma(\frac{\Delta-2}{2})\Gamma(\frac{\Delta+1}{2})} & \Delta > 2, \ J = 1, \\[2ex] -\frac{2^{2\Delta-1}(\Delta^2-1)\Gamma(\Delta-\frac{1}{2})}{\sqrt{\pi}(\Delta-2)^2\Delta\Gamma(\Delta-3)} & \Delta > 3, \ J = 2, \\[2ex] \qquad \cdots \end{cases}
\tag{3.22}
$$

for non-conserved operators. Note for conserved currents in odd dimensions, the function $E^{(3)}[\mathcal{O}]$ may depend on explicit choice of the cutoff. For example, a transformation $\epsilon \to \epsilon(1+a\epsilon)$ may shift its value. This is because the degree is 0, there is no logarithmic divergence at all.

## 3.3 UV/IR relation

The UV/IR relation (3.16) relates type-$(m)$ CCF to type-$(m-1,1)$ CCF. This relation may simplify computation in many cases. To see this point, let's compute the following type-$(2)$ CCF in two dimensions

$$
\begin{aligned}
\langle Q_A[\mathcal{O}]^2 \rangle_c &= \int_{-1}^{1} dz_1 \int_{-1}^{1} dz_2 \frac{(1-z_1^2)^{h-1}(1-z_2^2)^{h-1}}{(z_1-z_2)^{2h}} \\
&= \frac{(-1)^{-h}\sqrt{\pi}\Gamma(h)}{\Gamma(h+\frac{1}{2})} \int_{-1}^{1} dz_1 \frac{1}{1-z_1^2} \\
&= \frac{(-1)^{-h}\sqrt{\pi}\Gamma(h)}{\Gamma(h+\frac{1}{2})} \log\frac{2}{\epsilon}.
\end{aligned}
\tag{3.23}
$$

This is a double integral with poles at $z_1 = z_2$. We regularize the integral by ignoring these poles at the second step. At the last step, we insert a UV cutoff to regularize the integral. However, using UV/IR relation, one just need to fix the coefficient $D$ which is related to the large distance behaviour of the type-$(1,1)$ CCF,

$$
\langle Q_A[\mathcal{O}]Q_B[\mathcal{O}] \rangle_c = \int_{-1}^{1} dz_1 \int_{-1}^{1} dz_2 \frac{(1-z_1^2)^{h-1}(1-z_2^2)^{h-1}}{(z_1-z_2+x_0)^{2h}}.
\tag{3.24}
$$

In the large distance limit, $x_0 \to \infty$, the integral becomes simpler

$$
\begin{aligned}
\langle Q_A[\mathcal{O}]Q_B[\mathcal{O}] \rangle_c &\approx \int_{-1}^{1} dz_1 \int_{-1}^{1} dz_2 \frac{(1-z_1^2)^{h-1}(1-z_2^2)^{h-1}}{x_0^{2h}} \\
&= 4^{-h}\left(\frac{\sqrt{\pi}\Gamma(h)}{\Gamma\left(h+\frac{1}{2}\right)}\right)^2 \eta^h.
\end{aligned}
\tag{3.25}
$$

We have used the relation $\eta \approx \frac{4}{x_0^2}$ in the large distance limit. Then we can read out

$$
D^{(2)}[\mathcal{O},\mathcal{O}] = (-1)^{-h} 4^{-h}\left(\frac{\sqrt{\pi}\Gamma(h)}{\Gamma\left(h+\frac{1}{2}\right)}\right)^2.
\tag{3.26}
$$

Combining UV/IR relation and (3.18), we find

$$
p_1^{(2)}[\mathcal{O},\mathcal{O}] = E^{(2)}[\mathcal{O}] \times D^{(2)}[\mathcal{O},\mathcal{O}] = \frac{(-1)^{-h}\sqrt{\pi}\Gamma(h)}{\Gamma(h+\frac{1}{2})}.
\tag{3.27}
$$

The result is exactly the same as (3.23). We use the UV/IR relation to obtain type-(3) CCF for type-J OPE blocks in two dimensions, the cutoff independent coefficient is

$$p_1^{(2)}[\mathcal{O}_1, \mathcal{O}_2, \mathcal{O}_3] = \frac{C_{123}\pi^{3/2}(-1)^{\frac{h_1+h_2+h_3}{2}}\Gamma(h_1)\Gamma(h_2)\Gamma(h_3)\kappa}{\Gamma(\frac{1+h_1+h_2-h_3}{2})\Gamma(\frac{1+h_1+h_3-h_2}{2})\Gamma(\frac{1+h_2+h_3-h_1}{2})\Gamma(\frac{h_1+h_2+h_3}{2})}, \tag{3.28}$$

where the constant $\kappa = \frac{1}{2}[1 + (-1)^{h_1+h_2+h_3}]$. We notice that the result is totally symmetric under the exchange of any two conformal weights. This is a cyclic identity for $m = 3$

$$p_q^{(d)}[\mathcal{O}_1, \mathcal{O}_2, \mathcal{O}_3] = p_q^{(d)}[\mathcal{O}_2, \mathcal{O}_3, \mathcal{O}_1] = p_q^{(d)}[\mathcal{O}_3, \mathcal{O}_1, \mathcal{O}_2]. \tag{3.29}$$

Note that the cyclic identity cannot be assumed to be a priori since we are dealing with the limits of rather different quantities. However, interestingly, the UV/IR relation and the cyclic identity could be checked for all the examples in the following. For four dimensional type-($m$) CCF (m=2,3), we list the cutoff independent coefficients below [17].

- Type-(2). The normalization constants are set to 1.

    - Spin 1-1 conserved currents.

    $$p_1^{(4)}[\mathcal{J}_\mu, \mathcal{J}_\nu] = -\frac{\pi^2}{3}. \tag{3.30}$$

    - Spin 2-2 conserved currents.

    $$p_1^{(4)}[T_{\mu\nu}, T_{\rho\sigma}] = -\frac{\pi^2}{40}. \tag{3.31}$$

    - Spin 0-0 non-conserved operators.

    $$p_2^{(4)}[\mathcal{O}, \mathcal{O}] = -\frac{4\pi^2(\Delta-1)\Gamma(\Delta-2)^2\Gamma(\frac{\Delta}{2})^4}{\Gamma(\Delta)^2\Gamma(\Delta-1)^2}. \tag{3.32}$$

    - Spin 1-1 non-conserved operators.

    $$p_2^{(4)}[\mathcal{O}_\mu, \mathcal{O}_\nu] = -\frac{4^{1-\Delta}\pi^3\Delta\Gamma(\frac{\Delta-3}{2})\Gamma(\frac{\Delta+1}{2})}{\Gamma(\frac{\Delta}{2}+1)^2}, \quad \Delta > 3. \tag{3.33}$$

    - Spin 2-2 non-conserved operators.

    $$p_2^{(4)}[\mathcal{O}_{\mu\nu}, \mathcal{O}_{\rho\sigma}] = -\frac{3\pi^2(\Delta-2)\Delta^2\Gamma(\frac{\Delta}{2}-2)^2\Gamma(\frac{\Delta}{2}-1)^2}{64\Gamma(\Delta-4)\Gamma(\Delta+2)}, \quad \Delta > 4. \tag{3.34}$$

- Type-(3).

    - Spin 1-1-2 conserved currents. The three point function of zero components are fixed by conformal symmetry

    $$\langle T_{00}(x_1)\mathcal{J}_0(x_2)\mathcal{J}_0(x_3)\rangle_c = \frac{C_{T\mathcal{J}\mathcal{J}}}{x_{12}^4 x_{13}^2 x_{23}^2}. \tag{3.35}$$

    Then the coefficient

    $$p_1^{(4)}[\mathcal{J}_\mu, \mathcal{J}_\nu, T_{\rho\sigma}] = -\frac{\pi^3}{2}C_{T\mathcal{J}\mathcal{J}}. \tag{3.36}$$

– Spin 2-2-2 conserved currents. The three point function of zero components are fixed by conformal symmetry

$$\langle T_{00}(x_1)T_{00}(x_2)T_{00}(x_3)\rangle_c = \frac{C_{TTT}}{x_{12}^4 x_{13}^4 x_{23}^4}. \tag{3.37}$$

Then the coefficient

$$p_1^{(4)}[T_{\mu\nu}, T_{\rho\sigma}, T_{\alpha\beta}] = \frac{\pi^3}{12} C_{TTT}. \tag{3.38}$$

– Spin 0-0-0 non-conserved currents.

$$
\begin{aligned}
&p_2^{(4)}[\mathcal{O}_1, \mathcal{O}_2, \mathcal{O}_3] \\
=\ & -2^{4-\Delta_1-\Delta_2-\Delta_3}\pi^3 C_{123} \int_{\mathbb{D}^2} d\zeta d\bar\zeta (\zeta+\bar\zeta)^2 \int_{\mathbb{D}^2} d\zeta' d\bar\zeta'(\zeta'+\bar\zeta')^2 \\
&\times (1-\zeta^2)^{\frac{\Delta_1-4}{2}}(1-\bar\zeta^2)^{\frac{\Delta_1-4}{2}}(1-\zeta'^2)^{\frac{\Delta_2-4}{2}}(1-\bar\zeta'^2)^{\frac{\Delta_2-4}{2}} \\
&\times \int_0^\pi d\theta \frac{\sin\theta}{(a+b\cos\theta)^{\frac{\Delta_{12,3}}{2}}}.
\end{aligned} \tag{3.39}
$$

Though the expression (3.39) is not symmetric superficially under the exchange of any two conformal weights, we checked explicitly that it satisfies the cyclic identity for integer conformal weights. There could be singularities when $\zeta, \bar\zeta, \zeta', \bar\zeta'$ are close to the boundary $-1$ and $1$, we can deal with these singularities for integer conformal weights explicitly. We don't find a straightforward way to regularize the integral for non-integer conformal weights. It is interesting to find an unambiguous way to define $p_q$ for general operators.

For $m = 4$, the cyclic identities are much more harder to check. We considered type-(4) CCF for massless free scalar theory [13, 14]. In this theory, one can construct an infinite tower of conserved currents with even spin [29]. The four point functions can be calculated explicitly. Therefore we can find type-(3, 1) and type-(4) CCFs and read out the corresponding coefficients. For example, for spin-2-2-2-4 conserved currents [14],

$$D[2,2,2,4] = \frac{3}{70}D[2,2,4,2]. \tag{3.40}$$

Both of them lead to the cutoff independent coefficients

$$p_1^{(2)}[2,2,2,4] = \frac{2\Gamma(8)}{\Gamma(4)^2}D[2,2,2,4] = \frac{2\Gamma(4)}{\Gamma(2)^2}D[2,2,4,2] = p_1^{(2)}[2,2,4,2]. \tag{3.41}$$

The cyclic identity is obeyed.

## 3.4 Discussion

The UV/IR relation should be slightly modified when the CCF contains both type-J and type-O OPE blocks. One simple example is the following type-(3) CCF

$$\langle Q_A[\mathcal{J}]Q_A[\mathcal{O}]Q_A[\tilde{\mathcal{O}}]\rangle_c, \tag{3.42}$$

where $Q_A[\mathcal{J}]$ is a type-J OPE block while $Q_A[\mathcal{O}]$ and $Q_A[\tilde{\mathcal{O}}]$ are type-O OPE blocks. This CCF is related to the following two type-(2, 1) CCFs

$$\langle Q_A[\tilde{\mathcal{O}}]Q_A[\mathcal{J}]Q_B[\mathcal{O}]\rangle_c = D^{(d)}[\tilde{\mathcal{O}}, \mathcal{J}, \mathcal{O}]G_{\Delta,J}^{(d)}(\xi), \tag{3.43}$$

$$\langle Q_A[\mathcal{O}]Q_A[\tilde{\mathcal{O}}]Q_B[\mathcal{J}]\rangle_c = D^{(d)}[\mathcal{O}, \tilde{\mathcal{O}}, \mathcal{J}]G_{\Delta',J'}^{(d)}(\xi). \tag{3.44}$$

We choose $d = 4$. Taking the limit $A \to B$ from (3.43), we find a type-(3) CCF with degree $q = 2$, the UV/IR relation reads

$$p_2^{(4)}[\tilde{\mathcal{O}}, \mathcal{J}, \mathcal{O}] = E^{(4)}[\mathcal{O}] \times D^{(4)}[\tilde{\mathcal{O}}, \mathcal{J}, \mathcal{O}]. \tag{3.45}$$

We can also take the limit $A \to B$ from (3.44), then we will find a type-(3) CCF with degree $q = 1$, the UV/IR relation reads

$$p_1^{(4)}[\mathcal{O}, \tilde{\mathcal{O}}, \mathcal{J}] = E^{(4)}[\mathcal{J}] \times D^{(4)}[\mathcal{O}, \tilde{\mathcal{O}}, \mathcal{J}]. \tag{3.46}$$

The equations (3.45) and (3.46) are not identical superficially since the subscript $q$ are not equal to each other. However, an explicit calculation for spin 2-0-0 and spin 2-2-0 in four dimensions [17] shows that the coefficient $D^{(4)}[\mathcal{O}, \tilde{\mathcal{O}}, \mathcal{J}]$ is actually divergent logarithmically,

$$D^{(4)}[\mathcal{O}, \tilde{\mathcal{O}}, \mathcal{J}] = D_{\log}^{(4)}[\mathcal{O}, \tilde{\mathcal{O}}, \mathcal{J}] \log \frac{R}{\epsilon} + \cdots. \tag{3.47}$$

The terms in $\cdots$ are finite and depends on cutoff scale. Due to the logarithmic divergence behaviour of the coefficient $D^{(4)}[\mathcal{O}, \tilde{\mathcal{O}}, \mathcal{J}]$, the degree of type-(3) CCF from (3.44) increases by 1, the modified UV/IR relation becomes

$$p_2^{(4)}[\mathcal{O}, \tilde{\mathcal{O}}, \mathcal{J}] = E^{(4)}[\mathcal{J}] \times D_{\log}^{(4)}[\mathcal{O}, \tilde{\mathcal{O}}, \mathcal{J}]. \tag{3.48}$$

We checked explicitly that the two constants (3.45) and (3.48) are equal to each other. The cyclic identity is still satisfied after counting the logarithmic divergence of the $D$ function.

## 4 Generalizations

The area law and logarithmic behaviour in the subleading terms can be extended in different directions. In this section, we mention several extensions.

- UV/IR relation. In general, one can uplift any type-($m$) CCF to a type-($p, m-p$) CCF

$$\langle Q_A[\mathcal{O}_1] \cdots Q_A[\mathcal{O}_m] \rangle_c \xrightarrow{uplift} \langle Q_A[\mathcal{O}_1] \cdots Q_A[\mathcal{O}_p] Q_B[\mathcal{O}_{p+1}] \cdots Q_B[\mathcal{O}_m] \rangle_c, \quad 1 \le p \le m-1. \tag{4.1}$$

  When $p$ is not 1 or $m - 1$, the type-($p, m-p$) CCF is not a conformal block. It is still a function of cross ratio $\xi$, therefore it should reproduce the type-($m$) CCF after taking the limit $A \to B$,

$$\langle Q_A[\mathcal{O}_1] \cdots Q_A[\mathcal{O}_m] \rangle_c = \lim_{\xi \to -\infty} \langle Q_A[\mathcal{O}_1] \cdots Q_A[\mathcal{O}_p] Q_B[\mathcal{O}_{p+1}] \cdots Q_B[\mathcal{O}_m] \rangle_c. \tag{4.2}$$

  Obviously, this also defines a UV/IR relation between $p_q^{(d)}$ and several coefficients in the type-($p, m-p$) CCF. Since the right hand side is not proportional to any conformal block, it is not easy to write out an explicit formula. Nevertheless, one may still check the relation (4.2) case by case. One example is to consider the type-($2, 2$) CCF of the modular Hamiltonian in CFT$_2$. By making use of the universal feature of the CCF of the stress tensor, one can fix the generator of type-($m_1, m_2$) CCFs [14]

$$T_{A \cup B}(\mu_1, \mu_2) = -\frac{c}{2} \operatorname{tr} \log[\mathbf{1} - \begin{pmatrix} \mathcal{A} & \mathcal{C} \\ \mathcal{D} & \mathcal{B} \end{pmatrix}], \tag{4.3}$$

where the matrices $\mathcal{A}, \mathcal{B}, \mathcal{C}$ and $\mathcal{D}$ are

$$\mathcal{A}_{xx'} = \frac{\eta^2}{4} \int_0^\infty dy \frac{\sqrt{xx'}y \, \sinh \pi\mu_1 x \, \sinh \pi\mu_2 y}{\sinh \pi x' \, \sinh \pi y \, \sinh \pi(1+\mu_1)x \, \sinh \pi(1+\mu_2)y} \left(\frac{x_{13}}{x_{23}}\right)^{i(x-x')} \mathcal{F}(x, x', y),$$
(4.4)

$$\mathcal{B}_{xx'} = \frac{\eta^2}{4} \int_0^\infty dy \frac{\sqrt{xx'}y \, \sinh \pi\mu_1 x \, \sinh \pi\mu_2 y}{\sinh \pi x' \, \sinh \pi y \, \sinh \pi(1+\mu_1)x \, \sinh \pi(1+\mu_2)y} \left(\frac{x_{13}}{x_{23}}\right)^{-i(x-x')} \mathcal{F}(x', x, y),$$
(4.5)

$$\mathcal{C}_{xx'} = \frac{\eta^2}{4} \int_0^\infty dy \frac{\sqrt{xx'}y \, \sinh \pi\mu_1 x \, \sinh \pi\mu_2 y}{\sinh \pi x' \, \sinh \pi y \, \sinh \pi(1+\mu_1)x \, \sinh \pi(1+\mu_2)y} \left(\frac{x_{13}}{x_{23}}\right)^{i(x+x')} \mathcal{F}(x, -x', y),$$
(4.6)

$$\mathcal{D}_{xx'} = \frac{\eta^2}{4} \int_0^\infty dy \frac{\sqrt{xx'}y \, \sinh \pi\mu_1 x \, \sinh \pi\mu_2 y}{\sinh \pi x' \, \sinh \pi y \, \sinh \pi(1+\mu_1)x \, \sinh \pi(1+\mu_2)y} \left(\frac{x_{13}}{x_{23}}\right)^{-i(x+x')} \mathcal{F}(-x, x', y),$$
(4.7)

with

$$\begin{aligned}
\mathcal{F}(x, x', y) &= {}_2F_1(1+ix, 1-iy, 2, -\eta) \, {}_2F_1(1-ix', 1+iy, 2, -\eta) \\
&+ {}_2F_1(1+ix, 1+iy, 2, -\eta) \, {}_2F_1(1-ix', 1-iy, 2, -\eta).
\end{aligned}$$
(4.8)

$\mathcal{F}$ and its complex conjugate obey

$$\mathcal{F}^*(x, x', y) = \mathcal{F}(x', x, y), \quad \mathcal{F}^*(-x, -x', y) = \mathcal{F}(x, x', y),$$
(4.9)

so

$$\mathcal{A} = \mathcal{B}^*, \quad \mathcal{C} = \mathcal{D}^*.$$
(4.10)

We read out the first few CCFs

$$\begin{aligned}
\langle H_A^m \rangle_c &= \frac{cm!}{12} \log \frac{2}{\epsilon}, \\
\langle H_A^{m-1} H_B \rangle_c &= \frac{cm!}{144} G_2^{(2)}(\eta), \\
\langle H_A^2 H_B^2 \rangle_c &= c\Big\{ \frac{1+\eta}{\eta^2} \Big[ 4\text{Li}_3(1+\eta) - 2\log(1+\eta)\text{Li}_2(1+\eta) + \frac{2\log(1+\eta)}{3}\text{Li}_2(-\eta) - 4\zeta(3) \\
&\quad + \frac{1+\eta}{3}\log^2(1+\eta) - \frac{\pi^2}{3}\log(1+\eta) \Big] + \frac{2+\eta}{3\eta}\big[ 2\text{Li}_2(-\eta) + 3\log(1+\eta) \big] - \frac{4}{3} \Big\},
\end{aligned}$$
(4.11)

where the polylogrithm $\text{Li}_n(z)$ is

$$\text{Li}_n(z) = \sum_{k=1}^\infty \frac{z^k}{k^n}.$$
(4.12)

The relation (4.2) can be checked for $p = 2, m = 4$. The right hand side is

$$\lim_{\eta \to -1} \langle H_A^2 H_B^2 \rangle_c = 2c \log \frac{2}{\epsilon} + \cdots.$$
(4.13)

The cutoff independent coefficient $2c$ matches with the one in $\langle H_A^4 \rangle_c$.

- New power law. In the previous discussion, we focus on the case that $B$ and $A$ coincide with each other. However, there are other cases that the CCFs are still divergent. One can consider the limit that $A$ just attaches the edge of $B$,

$$R' = 1, \quad x_0 = 2 + \epsilon, \quad \epsilon \to 0. \tag{4.14}$$

The cross ratio $\xi$ does not approach $-\infty$ but 1

$$\xi = \frac{4}{(2+\epsilon)^2} = 1 - \epsilon + \cdots. \tag{4.15}$$

We can define a new CCF which is also divergent from type-$(m-1,1)$ CCF

$$\langle Q_A[\mathcal{O}_1]\cdots Q_A[\mathcal{O}_{m-1}] \odot Q_B[\mathcal{O}_m]\rangle_c = \lim_{\xi \to 1}\langle Q_A[\mathcal{O}_1]\cdots Q_A[\mathcal{O}_{m-1}]Q_B[\mathcal{O}_m]\rangle_c. \tag{4.16}$$

The continuation of conformal block tells us that the new CCF obeys a new power law

$$\langle Q_A[\mathcal{O}_1]\cdots Q_A[\mathcal{O}_{m-1}] \odot Q_B[\mathcal{O}_m]\rangle_c = \bar{\gamma}(\frac{R}{\epsilon})^{\frac{d-2}{2}} + \cdots + \bar{p}_q^{(d)}\log^q\frac{R}{\epsilon} + \cdots. \tag{4.17}$$

The leading term is proportional to

$$\mathcal{L} = R^{\frac{d-2}{2}} = \sqrt{\mathcal{A}}, \tag{4.18}$$

which is the characteristic length of the region $A$ in four dimensions. In two dimensions, the leading term is a logarithmic term with power $q$. In this case, there is a new UV/IR relation between $\bar{p}_q$ and $D$ coefficient , we write it schematically

$$\bar{p}_q = \bar{E} \times D. \tag{4.19}$$

The function $\bar{E}^{(d)}[\mathcal{O}]$ is proportional to $E^{(d)}[\mathcal{O}]$. The proportional constant is shown below.

- $d$ is even.

  For conserved current $\mathcal{O}$ with conformal weight $\Delta = J + d - 2$,

  $$\bar{E}^{(d)}[\mathcal{O}] = \frac{(-1)^J}{2}E^{(d)}[\mathcal{O}]. \tag{4.20}$$

  For non-conserved current $\mathcal{O}$ with conformal weight $\Delta$ and spin $J$,

  $$\bar{E}^{(d)}[\mathcal{O}] = \frac{(-1)^J}{4}E^{(d)}[\mathcal{O}]. \tag{4.21}$$

  We checked the relation for $d = 2, 4$ and spin $J \leq 2$.

- $d$ is odd.

  For non-conserved current $\mathcal{O}$ with conformal weight $\Delta$ and spin $J$,

  $$\bar{E}^{(d)}[\mathcal{O}] = \frac{(-1)^J}{2}E^{(d)}[\mathcal{O}]. \tag{4.22}$$

  For conserved current $\mathcal{O}$, there is no logarithmic divergent term in the CCF.

  We checked the relation for $d = 3$ and spin $J \leq 2$.

Since $D$ function is the same, we find a relation between two cutoff independent coefficients $p$ and $\bar{p}$,

$$\frac{p}{E} = \frac{\bar{p}}{\bar{E}}. \tag{4.23}$$

# 5 Summary and outlook

In this report, we have introduced the area law (3.1) of type-($m$) CCFs of OPE blocks. It is a generalization of the area law of entanglement entropy. We will list several open problems for future work.

- Higher $m \geq 4$. In most of the work, we consider type-(2) and type-(3) CCFs. This is because the structure of $m$-point correlation function of primary operators in CFT is fixed up to $m = 3$. For $m = 4$, we can also extract cutoff independent information for two dimensional massless free scalar theory [16].

- UV/IR relation. The UV/IR relation

$$p = E \times D \tag{5.1}$$

  has been checked for several examples. A rigorous proof is still lacking.

- Cyclic identity. The cyclic identity of $p$ reflects the fact that $p$ is independent of the way to regularize the type-($m$) CCF. However, we feel that a direct computation is impossible to check this identity.

- New power law. We generalize the type-($m_1, m_2$) CCF to the case that $A$ and $B$ just attach to each other. The corresponding CCF is divergent with a new power law (4.17). The corresponding new UV/IR relation

$$\bar{p} = \bar{E} \times D \tag{5.2}$$

  also needs understanding.

- Deformed reduced density matrix. This exponential operator is similar to the "Wilson loop" in gauge theories [30,31] despite the fact that the OPE block has no lower bound in general. When the OPE block has a lower bound, the logarithm of the vacuum expectation value of the deformed reduced density matrix

$$\log \langle e^{-\mu Q_A} \rangle \tag{5.3}$$

  should also obey area law with logarithmic divergence. There may be a gravitational dual for this quantity as [32,33]. The similarity of the area law between this program and black hole entropy implies that the classical part contributes to the area term while quantum effects lead to logarithmic corrections.

- Multiple integrals. According to the method of continuation of conformal block, area law of type-($m$) CCF is protected by conformal invariance. However, the method of continuation itself cannot guarantee that it always leads to the correct result. One has to develop other methods to deal with the multiple integrals. In two dimensions, one should generalize Selberg integrals [34,35] to include more parameters [16].

# Acknowledgements

This work was supported by NSFC Grant No. 12005069.

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
