# Peer review of "Area law and OPE blocks in conformal field theory"

_SciPost Physics Proceedings, doi:SciPost Phys. Proc. 4, 013 (2021)_

## Round 1 · Referee Report · Anonymous (Referee 1) · 2020-12-22

Strengths

1- Clear review of background material

2- Technical results seem correct and important to understand the physical interpretation of the generalizations of the reduced density matrix operator described in the draft.

Weaknesses

1- It is not clear why the exponential operators studied in the paper are important. It seems the calculations have been done just because they can be performed.

2- All of the results involve at most correlators involving three operators. All results are thus fixed by conformal invariance, which makes the calculations almost trivial.

Report

In this paper, the author motivates the study of a specific set of conformal correlators by introducing exponential operators that generalize the usual reduced density matrix of a CFT in a ball shaped region.

The results concern the connected correlator of OPE blocks that generalize the construction of the modular Hamiltonian in a conformal field theory.
Explicit results are given for d=2,3,4, and 6.
The result is a cut-off independent coefficient in the log-term divergence of the correlator, and it is obtained by explicitly computing the correlators using known conformal block expressions.

In its current state I am not comfortable recommending the draft for publication. Even though the technical results of the paper seem correct and might prove important in the future, I do not think the calculations are justified physically.

Requested changes

1- I recommend that a physical interpretation of equation 2.29 is formulated. The objects studied in the paper are exponential operators constructed from OPE blocks. They are denoted by \rho_A. (this is confusing given that \rho_A is how the reduced density matrix is also denoted in the paper.

It is not clear why this object is interesting in any way. It is not clear whether it can be interpreted as a reduced density matrix. The author mentions this issue in the discussion, giving the impression that the calculations have been performed simply because they can be done, but there does not seem to be much physics justifying the work.

2 - All of the results in the paper involve at most m=3, which implies that the correlators playing a role in the calculation are fully fixed by conformal invariance. This either should be expanded on with more cases, or should be mentioned in the abstract, as currently there seems to be a disconnect between what is implied in the abstract/introduction, and the actual scope of the paper.

3 - (Minor issue). There are problems with the grammar and vocabulary in the paper. Many indefinite articles are missing, and some words are used incorrectly (evolution vs evaluation for example).

  • validity: high
  • significance: low
  • originality: low
  • clarity: top
  • formatting: acceptable
  • grammar: acceptable

Author:  Jiang Long  on 2020-12-30  [id 1118]

(in reply to Report 1 on 2020-12-22)

  1. The referee writes: " I recommend that a physical interpretation of equation 2.29 is formulated. The objects studied in the paper are exponential operators constructed from OPE blocks. They are denoted by \rho_A. (this is confusing given that \rho_A is how the reduced density matrix is also denoted in the paper. It is not clear why this object is interesting in any way. It is not clear whether it can be interpreted as a reduced density matrix. The author mentions this issue in the discussion, giving the impression that the calculations have been performed simply because they can be done, but there does not seem to be much physics justifying the work."

Our response: In the new manuscript, we added reference [15] below equation 2.29 as an interpretation of the exponential object discussed in this paper. Our definition of the deformed reduced density matrix is a direct generalization of the operator in the context of charged Renyi entropy.

  1. The referee writes: "All of the results in the paper involve at most m=3, which implies that the correlators playing a role in the calculation are fully fixed by conformal invariance. This either should be expanded on with more cases, or should be mentioned in the abstract, as currently there seems to be a disconnect between what is implied in the abstract/introduction, and the actual scope of the paper. "

Our response: We discussed several cases for m=4 in two dimensions in the paragraph above equation (3.40). We could check all the points in this paper for two dimensional massless free scalar theory.

  1. The referee writes: "There are problems with the grammar and vocabulary in the paper. Many indefinite articles are missing, and some words are used incorrectly (evolution vs evaluation for example)."

Our response: We improved the manuscript on the grammar and vocabulary in the new version.

---

## Round 2 · Referee Report · Anonymous (Referee 2) · 2021-2-10

Strengths

1- Contains concrete computations and results that encode nontrivial information of CFT.

Weaknesses

1- Some key quantities and major steps lack clear definition/explanation. 2- Overall physical significance is unclear. 3- Exposition, grammar, and clarity of writing needs major improvement.

Report

The author studies nonlocal operators in CFT known as "OPE blocks", computes their correlators, and extracts the universal logarithmic pieces in the small cutoff limit. The key results are a UV/IR relation (3.11) and a cyclic identity (3.29). However, I have some questions about both results, and about the significance of "area law" in this story. I have listed my requested changes in order of appearance in the paper. The more important ones are 5, 13, 14, 15, 16, 18, 19.

Overall, I think the paper can be improved by making a stronger case for why the studied quantity may be physically interesting (point 19), and/or computing more examples (point 18). I cannot yet recommend this paper for publication until a major revision is made, possibly with some enrichment of the content.

Disclaimer: I am an expert in CFT but not specifically in OPE blocks.

Requested changes

1- line 25: Is OPE block a new topic? Maybe its detailed study as a non-local operator, its relation to information theory, or its application in the holographic context is, but references [5,6] are certainly not new. I suggest the author be more specific about what the new topic is.

2- line 33: Leading order in what?

3- line 39: I cannot understand the sentence: "This leads to the conjecture that OPE block may be related to area law as modular Hamiltonian."

4- line 44: Natural numbers do not include 0, but as noted later type-J in 3d has q=0. It is also weird to not just say "0, 1, 2", instead of "natural number no larger than 2". Also, is it clear that the expansion cannot have fractional powers of the log?

5- line 65: The author defines the "area law" of a quantity Q in a(A) in its small cutoff expansion as the leading non-universal piece being proportional to the area of A, but the key quantity of interest is actually a subleading logarithmic piece. Thus, the main slogan "area law <-> OPE block" appears a little inaccurate.

6- line 67: Normally when we say QFT, it does not include Einstein gravity as an example.

7- (2.7) and line 214: I suppose ... also contains other log pieces with smaller power, but their coefficients are not universal (say under rescaling $\epsilon \to 2\epsilon$). Maybe a slight clarification like this is helpful.

8- line 89: Calling J the "magnitude" might be a little confusing, since $\sum_{i,j} J_{ij}^2 \neq J^2$.

9- line 109: Does "the same" mean the same quantum numbers or literally the same operator?

10- line 141: I do not understand what c are, and why they are free parameters. From (2.12) and (2.14), Q is unambiguously defined if one fixes the normalization of O. Is c just the normalization constant for O?

11- (2.26): Does $\gamma$ depend on n?

12- lines 151 and 210: When did R=1 happen?

13- line 176: As noted, the unboundedness of the OPE block makes (2.30) ill-defined. The thing that is actually well-defined is the CCF which you compute. Do you expect to be able to resum the CCFs to recover (2.30)? For instance, can you do this for $Q_A[T]$ to define non-integer Renyi?

14- (2.47): What is D? Are you calling G the conformal block, or DG the conformal block? Are you talking about the 4pt conformal block or some other conformal block? Just knowing that a function is an eigenfunction of the conformal Casimir does not completely specify the conformal block, in addition, boundary conditions/singularities must be specified. Is D kinematical or dynamical? Conformal blocks should be purely kinematical. Please add more explanation of D, since it enters one of the major results, the UV/IR relation (3.11), of the paper.

15- line 293-295: I do not understand the argument here. Just because a function has a symmetry in a certain limit does not mean it has a symmetry away from the limit. Please clarify.

16- line 313: Do you expect the cyclic identity to be only true for integer weights? Or do you expect it to be true for non-integer weights, but you only checked (3.39) in the integer case? If the latter, why not check non-integers?

17- line 387: "region" seems like the wrong word.

18- line 389: Can you at least compute them in simple examples like free theory or generalized free fields (holographically dual to weakly coupled gravity)?

19- Could you add some thoughts about how the quantity (3.11) may be potentially interesting? Could it be monotonic under RG flows? Does it have some nice holographic interpretation? Does it shed light on the analytic structure of some well-defined version of (2.30)?

20- Significant improvement of the writing is necessary. I am listing some of the mistakes I noticed, but there are certainly more.

Misspellings: e.g. line 23 and 50: vari"ous" line 178: Tayl"o"r

Articles are missing: e.g. line 32: "the" reduced line 146: "the" Renyi

Singular/plural mistakes: e.g. line 24: area"s" line 63: degree"s" line 85: term(s) line 105: numbers"s" line 320: lead(s)

Others: line 46: far away "from" line 82: chosen "to be" line 113: "inserted into" -> "are inserted" line 125: "intersects" with the" t = 0 slice "at/in" a unit ball line 134: correspond"ing" line 218: necessar"ily" line 235: set"ting" line 238: logarithmic"ally" line 242: coincid"ing" line 336: increases "by" line 344: "and" -> "or" line 396: attach "to" (3.17): extra "log"

The explanation of quantities following certain formulae uses very fractured sentences, for example below (2.10).

Equation overflow in lines 55, (3.39), (4.11).

---

## Round 2 · Referee Report · Anonymous (Referee 1) · 2021-2-10

Strengths

1- Clear review of background material

2- Technical results seem correct and important to understand the physical interpretation of the generalizations of the reduced density matrix operator described in the draft.

Weaknesses

1- It is not clear why the exponential operators studied in the paper are important. It seems the calculations have been done just because they can be performed.

Report

In this paper, the author motivates the study of a specific set of conformal correlators by introducing exponential operators that generalize the usual reduced density matrix of a CFT in a ball shaped region.

The results concern the connected correlator of OPE blocks that generalize the construction of the modular Hamiltonian in a conformal field theory.
Explicit results are given for d=2,3,4, and 6.
The result is a cut-off independent coefficient in the log-term divergence of the correlator, and it is obtained by explicitly computing the correlators using known conformal block expressions.

After the changes made by the authors in the second version of the draft, I recommend the draft for publication.

Requested changes

(Minor issue). There are problems with the grammar and vocabulary in the paper. Many indefinite articles are missing.

---

## Round 2 · List of Changes

1. In the new manuscript, we added reference [15] below equation 2.29 as an interpretation of the exponential object discussed in this paper. Our definition of the deformed reduced density matrix is a direct generalization of the operator in the context of charged Renyi entropy.
  2. We re-organised the paragraph above equation (3.40), emphasising the m=4 case for two dimensional massless free scalar theory.
  3. We changed "evaluation" to "evolution" in page 2.

---

## Round 3 · Referee Report · Anonymous (Referee 2) · 2021-3-24

Strengths
1- Proposes new observables in CFT. 2-Presents concrete computations and results.
Weaknesses
1- Grammar and clarity of the exposition can be improved.
Report
The revised manuscript has addressed many of the earlier problems, but a few issues remain. My most serious question is point 12 below, as the author seems to suggest that the key quantity, the coefficient of the leading logarithm, is not unambiguously defined for general operators. The grammar and clarity of the exposition can still be improved, but do not present a major problem.
I would recommend the manuscript for publication after the issues are addressed.
Requested changes
1-line 26: I suggest changing to something like "...is a relatively unexplored topic in conformal field theory, though it has been defined and discussed at the early stages of conformal field theory."
2-line 42: Here I think it is important to state clearly what you mean by area law, so I suggest putting "area law" in quotation marks, and adding a footnote saying that your notion of area law includes subleading corrections, and you use the slogan "area law" following the convention of geometric entanglement entropy. (copied from lines 96-97).
3-lines 42-43: If I understand your statement correctly, you want to say "This leads to the conjecture that similar to the modular Hamiltonian, general OPE blocks exhibit area law."
4-lines 47-48: By "We don’t find fractional powers of the logarithm.", is the claim of q = 0, 1, 2 just from the specific examples you studied, or can it be argued from properties of the conformal block? If it is just from examples, I suggest writing: "In all examples we studied, we found q = 0, 1, 2, but in general we do not rule out the possibility of other values." If there is some general argument, please explain.
5-line 86: "Natural number" excludes q=0, but type-J in 3d has q=0.
6-lines 165 and 231: "inserted the radius R = 1" -> "explicitly restored the radius R that was previously set to 1".
7-(2.18) to line 155: I still fail to understand the purpose of c. I can't find where a canonical normalization for Q is given. So why can't c be removed completely?
8-line 226: This sentence seems to imply that for m>2, D is not related to the normalization of operators. If so, why?
9-(3.1) The universal constant $p_q$ in (3.1) depends on your operator normalization. How are you normalizing O?
10-(2.45)-(2.47): Going from (2.45) to (2.47) should involve some Ward identity that relates the quantum operator L^2 to an explicit differential operator in \eta. Could you provide some explanation or a reference for this step? If \eta is the cross ratio of four points, then this is a standard exercise in CFT, but here \eta is defined in terms of some spatial regions.
11-lines 314-316: I maintain my opinion that just because a function has a symmetry in a certain limit does not mean it has a symmetry away from the limit. Therefore I think the argument given in this sentence is not very sensible even as a heuristic.
12-lines 338-339: Are you saying that there is no unambiguous way to regularize the divergences for non-integer weights? Does this mean that there is no unambiguous way to define $p_q$ for general operators?
13-It seems to me that the UV/IR relation is robust and follows from conformal symmetry, whereas the cyclic identity is a conjecture. If so, whenever you say that you "check the UV/IR relation and the cyclic identity", I suggest that you separate the two, since checking a conjecture is morally different from checking a fact.
14-Exposition and grammar. Just to point out two examples:
line 29 is unnecessarily fractured. It could be combined into "It is a smeared operator which is generated from a so-called (quasi-)primary operator, and extends the study of local operators in CFT to non-local operators.
line 264 still contains "far away to".

---

## Round 3 · List of Changes

- Line 30-31 and 40, we added the discussion on OPE blocks , emphasising that OPE block provides a novel look at the modular Hamiltonian. This is also the motivation to relate OPE block to area law in this paper.
- Line 35-37: we rewrote the UV divergences of the entanglement entropy.
- Line 47-48: we wrote explicitly the possible values of the degree q. There is no fractional power of the logarithm.
- Line 72: we added a sentence to extend the area law to general field theory.
- Line 88-89: we discussed the possible logarithmic pieces with smaller power. They don't provide any universal information.
- Line 94-97: we added one comment on the relation between the area law and OPE blocks.
- Line 101: we changed "so(d-1) spin J_{ij} with magnitude J" to "so(d-1) spin J".
- Line 120-121: we changed the "the external operators are the same" to "the external operators have the same quantum numbers".
- Line 154-155: the constant c is related to normalization of the operator . This explains why we can set c=1.
- Eq (2.26): \gamma\to\gamma(n).
- Line 138: we added "(R=1)". 12.Line 196-198: we added the convergence problem of the summation of eq. (2.31).
- Line 221-226: we discussed the coefficient D and conformal block G.
- Line 317-319: the cyclic identity is a conjecture. We don't prove its validity, however, we can check it case by case. 15.Line 337-339: we added the discussion on the non-integer conformal weights.
- Line 412: we changed "we restrict to the region m\ge 3" to "we consider type-(2) and type-(3) CCFs".
- Line 414-415: we added the reference on the type-(4) CCF for free theory.
- We rewrote (3.39) and (4.11).
- We improved the writing of the paper.

---

## Round 4 · Referee Report · Anonymous (Referee 2) · 2021-6-29

Strengths
1- Proposes new observables in CFT. 2- Presents concrete computations and results.
Weaknesses
1- It is unclear whether the key quantity $p_q$ is unambiguously defined for general operators.
Report
I am unconvinced by the statement that "there should be an unambiguous way to define $p_q$ for general operators" on lines 335-336. The author does not propose any regularization scheme for general conformal weights, let alone showing that the result is independent of the regularization.
I would recommend the manuscript for publication after the remaining issues are addressed.
Requested changes
(2.18): In response to the author's reply about the meaning of the normalization constant $c$, while I understand the author's point, I personally find it very confusing. The author is saying that $Q_A$ has no predefined normalization, and one just chooses $c$ to suit the context. Due to this context-dependent $c$, $Q_A$ is not a linear functional of $O$. Besides, $Q_A$ is a scalar number, so without a predefined normalization, it has no unambiguous meaning. I personally think it would be much more satisfying if $c$ is set to be $2\pi$ for any $O$, in which case $Q_A$ is a linear functional.
lines 335-336: See report. I suggest that the author softens the statement or presents evidence for this claim.
I kindly ask the author to review the next revision more carefully, as new errors and typos have appeared in every past revision, e.g. for the current one,
line 223- cdots -> \cdots
line 337- identity -> identities

Anonymous on 2021-06-07 [id 1492]
comment to the Editors included

---

## Round 4 · Author Response

1) The referee says: I still fail to understand the purpose of c. I can't find where a canonical normalization for Q is given. So why can't c be removed completely?
Our response: the constant c can be removed in principle. However, sometimes the normalization of the operator O is given, then Q still has the freedom to choose its own normalization. For example, usually, the stress tensor in integral of the modular Hamiltonian is defined unambiguously, then one should choose c=2\pi such that tr_A \rho_A=1.
2) The referee says: This sentence seems to imply that for m>2, D is not related to the normalization of operators. If so, why?
Our response: I rewrote the sentence. D also depends on the normalization of the operators.
3) The referee says: The universal constant $p_q$ in (3.1) depends on your operator normalization. How are you normalizing O?
Our response: I agree the constant $p_q$ depends on the operator normalization. I don't choose a definite normalization when I state general results. The normalization doesn't affect the validity of my results.
4) The referee says: Are you saying that there is no unambiguous way to regularize the divergences for non-integer weights...
Our response: No. I rewrote the sentence. I can't regularize the integral by straightforward computation for non-integer weights, but this doesn't rule out the possibility that the constant $p_q$ is still defined unambiguously by other means.

---

## Round 4 · List of Changes

1-line 25-26: I have changed the sentence to "...is a relatively unexplored topic in conformal field theory, though it has been defined and discussed at the early stages of conformal field theory."
2-line 41-42: I put "area law" in quotation marks and add a footnote to explain it.
3-line 42-43: I changed the sentence as the referee suggested.
4-line 47-48: I changed the sentence to "In all examples we studied, we found q = 0, 1, 2, but in general we do not rule out the possibility of other values."
5-line 85: I changed "nature number" in the original version to "nonnegative number".
6-line 163 : I rewrote the sentence as " we have restored the radius R that was previously set to 1". There are similar modification in line 229-230.
7-line 223-224: I changed the sentence to "For $m\ge 2$, the coefficients $D^{(d)}[\mathcal{O}_1, cdots,\mathcal{O}_m]$ are related to the normalization of the primary operators. For any $m\ge 3$, it also contains dynamical information of the theory."
8-(3.1): I added a footnote 3 to address the point that $p_q$ also depends on the normalization of the operators.
9-(2.47): I added a footnote 2 to explain equation (2.47).
10-line 312: I deleted original argument on cyclic identity, just mention that we could read out a cyclic identity (3.29) from (3.28).
11-line 334-336: I rewrote a comment on the regularization of the integral for non-integer conformal weight.
12-line 337: I changed the sentence to "For m\ge 4, the cyclic identity are ..."
13: Iine 28-30: I changed the sentence as the referee suggested.
14: line 262: I change "far away to" to "far away from".

---

## Round 5 · Referee Report · Anonymous (Referee 2) · 2021-7-15

Strengths

1- Proposes new observables in CFT. 2- Presents concrete computations and results.

Weaknesses

1- It is unclear whether the key quantity $p_q$ is unambiguously defined for general operators.

Report

The author studied nonlocal operators in CFT known as "OPE blocks" that generalize the modular Hamiltonian, computed their correlators and extracted the universal leading logarithmic piece in the small cutoff limit. The key results were a UV/IR relation (3.16) and a cyclic identity (3.29). The former was proven by exploiting conformal symmetry, and the latter was conjectured based on evidence. The author carried out concrete calculations with suitable regularization to obtain $p_q$ for operators of integer weights, and left open whether there is an unambiguous way to define $p_q$ for general operators.

I recommend the manuscript for publication.

---

## Round 5 · List of Changes

1) The referee is still concerned with the normalization of Q, indeed it could change the value of p_q. In this paper, I already set it to be 1 in lines 152-153.

2) I soften my statement on the p_q in lines 335-336.

3) I changed cdots to \cdots in line 223 and identity to identities in line 337.

---

## Editorial Decision

published